# The Role of Microbiota in Liver Transplantation and Liver Transplantation-Related Biliary Complications

**DOI:** 10.3390/ijms24054841

**Published:** 2023-03-02

**Authors:** Ulrich Wirth, Tianxiao Jiang, Josefine Schardey, Katharina Kratz, Mingming Li, Malte Schirren, Florian Kühn, Alexandr Bazhin, Jens Werner, Markus Guba, Christian Schulz, Joachim Andrassy

**Affiliations:** 1Department of General, Visceral, and Transplant Surgery, Ludwig-Maximilians-University Munich, 81377 Munich, Germany; 2Department of Medicine II, University Hospital, Ludwig-Maximilians-University Munich, 81377 Munich, Germany

**Keywords:** liver transplantation, biliary microbiome, biliary complications, multi-drug resistant microbiota, ischemic-type biliary lesions

## Abstract

Liver transplantation as a treatment option for end-stage liver diseases is associated with a relevant risk for complications. On the one hand, immunological factors and associated chronic graft rejection are major causes of morbidity and carry an increased risk of mortality due to liver graft failure. On the other hand, infectious complications have a major impact on patient outcomes. In addition, abdominal or pulmonary infections, and biliary complications, including cholangitis, are common complications in patients after liver transplantation and can also be associated with a risk for mortality. Thereby, these patients already suffer from gut dysbiosis at the time of liver transplantation due to their severe underlying disease, causing end-stage liver failure. Despite an impaired gut-liver axis, repeated antibiotic therapies can cause major changes in the gut microbiome. Due to repeated biliary interventions, the biliary tract is often colonized by several bacteria with a high risk for multi-drug resistant germs causing local and systemic infections before and after liver transplantation. Growing evidence about the role of gut microbiota in the perioperative course and their impact on patient outcomes in liver transplantation is available. However, data about biliary microbiota and their impact on infectious and biliary complications are still sparse. In this comprehensive review, we compile the current evidence for the role of microbiome research in liver transplantation with a focus on biliary complications and infections due to multi-drug resistant germs.

## 1. Introduction

Liver transplantation (LT) is the only available treatment for patients with end-stage liver disease and acute liver failure with a definite long-term survival benefit since the 1960s [1]. Despite significant improvements both in surgical techniques as well as in postoperative medical care [2], biliary tract reconstruction remains a major source of short- and long-term morbidity. Biliary complications occur in up to 40% of patients after orthotopic liver transplantation [3,4]. Biliary leakage and bile duct strictures are the most common biliary complications [5,6,7,8]. Strictures can be classified as anastomotic or non-anastomotic according to their localization. Non-anastomotic intrahepatic strictures (NAS) are the most troublesome biliary complication and are either associated with hepatic artery thrombosis or can be referred to as “ischemic-type biliary lesions (ITBL)” [9]. Furthermore, biliary tract infections can occur due to ascending infections or due to microbial colonization already present at the time of transplantation [10]. Especially in patients with repeated biliary interventions and antibiotic therapies, the biliary tract can be colonized by a wide spectrum of potential pathogen microbiota [11,12,13]. However, despite pure colonization, infections, especially with multi-drug resistant microbiota, play a crucial role, as they can cause severe morbidity and are associated with an increased risk for mortality [14,15,16]. All these biliary complications chronically impair liver graft function and can lead to graft failure and the need for re-transplantation in some of the cases [9,17,18].

The human gastrointestinal tract harbors a complex and diverse population of microorganisms known as the gut microbiome. Given its anatomical position, the liver has a bidirectional relationship with the intestine and its microbiota, known as the “gut-liver axis,” which exhibits circular causality [19]. The liver thus represents the first line of defense against gut-derived antigens and toxicity factors. End-stage liver disease, as well as LT, is often associated with changes in the composition of the gut microbiome due to antibiotic therapy, interventions, altered anatomy from surgery, biliary complications, and the use of immunosuppression [20,21,22]. Available studies on humans have shown that LT can improve gut microbiota diversity in patients with end-stage liver diseases, accompanied by a higher relative abundance of beneficial bacteria and suppression of pathogenic Gram-negative bacteria [22], even with the use of immunosuppressants [23].

For a long time, the biliary tract has been considered a hostile territory for microbiota because bile acids, cholesterol, phospholipids, and biliverdin within the bile act as a biological detergent that emulsifies lipids and thus dissolves bacterial membranes. Various works on animal models and humans could disprove that the biliary tract is sterile [24,25,26,27]. Furthermore, a recent study on humans showed that the biliary tract seems to have a complex microbiota, even in healthy individuals [28]. Wu et al. even proposed that in humans, the bacterial diversity is higher in the biliary tract than in the intestine [29].

Due to bacterial colonization and immunosuppression, biliary infections are a frequent cause of biliary complications in liver transplant recipients [30,31]. Results from previous clinical studies have shown that pathogenic bacteria can be detected in specimens of bile or bile ducts in patients with biliary complications after liver transplantation [13,32].

In this review, we want to elucidate the role of microbiota in the context of liver transplantation, with a specific focus on the growing issue of multi-drug resistant microbiota. Finally, the biliary complications of LT are discussed in terms of the underlying role of the local microbial niche.

## 2. Gut-Liver Axis

The gut-liver axis refers to the bidirectional relationship between the intestine and the liver, in which they communicate and interact with each other through various pathways, including the portal venous and biliary systems (Figure 1). The liver plays a crucial role in detoxifying substances such as bacterial toxins from gut microbiota [19].

One of the major components of the gut-liver axis are bile acids, which serve as pleiotropic signaling molecules [33]. Primary bile acids are synthesized from cholesterol, which is later conjugated with glycine or taurine in the liver and released into the duodenum as primary bile salts. Bacteria in the gut will first deconjugate the primary bile salts into primary bile acids by removing the glycine or taurine and then modify the primary bile acids into secondary bile acids by 7α-dehydroxylation. Both primary and secondary bile acids could be further oxidized into less toxic oxo-bile acids or epimerized into non-toxic iso-bile acids by gut microbiota. While most bacteria can perform oxidization and synthesize oxo-bile acids, only some of the microbes are able to perform dehydroxylation and epimerization and synthesize iso-bile acids [33]. In patients with end-stage liver diseases, there is often a low bile acid secretion into the intestine as a result of cholestatic conditions [34]. Furthermore, there is a reduction in secondary bile acids due to reduced colonization by beneficial bacteria that perform the 7α-dehydroxylation [35]. The increase in secondary iso and oxo-bile acids in patients after LT serves as a biomarker of the proliferation in *Firmicutes* and reduction in *Proteobacteria*, which can be referred to as a reconstitution of a regular gut microbiome. Moreover, this provides better protection against opportunistic nosocomial bacterial overgrowth, such as *Clostridium difficile*, which are inhibited mainly by the secondary rather than the primary bile acids [36].

In healthy individuals, the gastrointestinal tract prevents bacterial translocation towards the liver through the intestinal barrier, mucus layer, and various antimicrobial proteins. A small amount of gut bacterial components could enter the portal blood circulation without triggering an immunological response due to the immune tolerance of the liver [37]. However, in conditions such as gut dysbiosis, inflammation, and loss of those barriers, an increased number of bacterial components enter the portal blood circulation, thereby triggering hepatic inflammation and fibrotic remodeling [38]. Seventy-five percent of primary sclerosing cholangitis (PSC) cases are associated with gut dysbiosis [39,40]. Furthermore, data demonstrate that colectomy due to various factors, including associated ulcerative colitis and colorectal cancer, can reduce PSC relapse in 37% of patients, reflecting the role of gut microbiota in inflammation-related liver diseases [41,42]. Not only for PSC but in other biliary diseases such as primary biliary cholangitis and biliary atresia, recent studies have demonstrated the essential role of innate immune responses [43,44]. These responses are triggered by microbial patterns, highlighting the importance of host-microbe interaction in the development of these conditions [37,38]. 

Overall, the gut-liver axis is important for the maintenance of immune function as well as metabolism and depends on the homeostasis of the gut microbiota. Therefore, it can be severely compromised in end-stage liver disease and associated gut dysbiosis.

## 3. Gut Microbiota in Chronic Liver Disease and Liver Transplantation

Gut microbial dysbiosis is associated with various liver diseases, including liver cirrhosis, hepatocellular carcinoma, and non-alcoholic fatty liver disease [45]. Therefore, LT is often associated with changes in the composition of the gut microbiome with possible restitution over time. At the time of LT, donor microbiota can be transferred to the recipient via the liver allograft [45]. Furthermore, increased intestinal permeability caused by surgery will allow certain pathogens to enter the portal or systemic circulation, and donor immune cells of the liver graft can interact with the recipient’s gut microbiome via the gut-liver axis [20,21,22]. A qPCR-based analysis of samples from 111 LT patients showed a decrease in *Bifidobacterium*, *Lactobacillus*, and *Faecalibacterum* and an increase in *Enterobacteriaceae* and *Enterococcus* in the gut microbiota of post-LT patients [46]. It is noteworthy that the indicators for the severity of liver cirrhosis, including a model for end-stage liver disease (MELD), Child-Pugh score, total bilirubin, pro-thrombin time test, international normalized ratio, creatinine level, and albumin level, were associated with higher amounts of a certain genus of bacteria, such as *Streptococcus, Veillonella* and *Clostridium* [47]. Patients with a higher amount of these bacteria had a significantly more severe illness compared to those with a low amount [47]. Another human study also showed a positive association between the severity of cirrhosis, as measured by Child-Pugh scores, and the presence of *Streptococcus* spp. [48]. These findings suggest that certain bacteria may play an active role in liver cirrhosis, as the correlation follows a “dose-response” pattern [47].

In another study, including 177 patients undergoing liver transplantation, Annavajhala et al. could demonstrate a correlation between disease etiology and gut microbiome diversity [16]. Especially patients with alcohol-related liver cirrhosis had significantly lower α-diversity measures compared to other diagnoses, whereas patients suffering from hepato-cellular carcinomas had significantly higher α-diversity measures [16]. Furthermore, the relative abundance of specific microbiota such as *Enterococcus casseliflavus*, *Veillonella dispar*, *Faecalibacterium prausnitzii,* or *Bifidobacterium bifidum* was different depending on disease etiology [16]. In their data, the microbial communities clustered differential for Child-Pugh A vs. Child-Pugh C patients as well as for patients with high or low MELD scores [16]. Specific changes in the gut microbiome could be detected in the post-LT phase, as in the first weeks following LT; usually, the diversity is reduced due to perioperative broad-spectrum antibiotic therapy [16]. In the perioperative phase, an enrichment in *Clostridiales*, *Streptococcus,* and *Enterococcus* spp. could be detected [16]. In the further months after LT, there was an increasing α-diversity, and distinct microbial patterns were identified in early (1–3 months) vs. late (6–12 months) post-LT phases [16]. Yet, there is not enough data that the gut or biliary microbiome is reliably able to predict outcomes or prognosis in LT patients [45].

Notably, LT and the administration of antibiotics can disrupt the microbial balance in the intestines of patients, leading to decreased beneficial and increased pathogenic bacteria [49]. Several studies on both animal models and humans report a beneficial effect of pro- and prebiotics on the outcome of liver transplantation regarding infectious complications, but no microbiome analysis has been included in these studies [50]. Taking into account that pre- and probiotics are proven to have beneficial effects on the gut microbiome, can enhance immune responses, and may have anti-inflammatory effects [51], probiotics and prebiotics may help reduce post-LT complications, including severe infections and liver injury, by altering the gut microbiome [50,52].

In general, moderate alterations of the gut microbiome after LT contribute to an increased tolerance of the liver allograft. Regulatory T cells (Tregs) secrete inhibitory cytokines and interact with CD80 that downregulates T cell activation, thereby inhibiting effector T cells [52]. Therefore, Tregs prevent the development of acute cellular rejection (ACR) by inducing a tolerogenic environment with an intact immune system in LT patients. Gut dysbiosis in post-LT patients increases abnormal portal circulation of bacterial products like LPS. Kupffer cells respond to such change by increasing concentrations of IL-10 with an anti-inflammatory effect [53]. In addition, such change also induces type 1 interferon and stimulates myeloid cell IL-10 production, thereby further increasing IL-10 concentrations in the liver [54]. Other research showed that LPS-induced local inflammation upregulates CD80 in a murine model [55]. If this finding is applicable to humans, post-LT dysbiosis could increase the apoptosis of CD8+ T cells and increase the tolerance of the liver allograft [56]. An increase in gut *Bacteroides fragilis* and *Bacteroides thetaiotaomicron* was found to drive regulatory T cell induction and differentiation in post-LT patients, which is correlated with a more tolerant alloimmune response [57]. However, excessive upregulation of Tregs could lead to reduced alloreactive T cell proliferation, thereby increasing the risk of post-LT malignancies and infections [45].

Gut dysbiosis can also affect the balance of CD4+ T cell subsets in mesenteric lymph nodes [58], and migrations of such altered T cells will promote hepatic injury or ACR [59]. In terms of post-LT hepatic ischemia-reperfusion (I/R) injury, several animal studies suggested that gut dysbiosis could exacerbate I/R injury-mediated development of early ACR in the post-LT patients because increased segmented filamentous bacteria could aid in IL-17 expression [60,61]. However, alterations in gut microbiota could potentially improve early ACR outcomes by alleviating hepatic I/R injury. It has been demonstrated that butyrate-altered gut microbiota can prevent NF-κB activation and upregulate 3,4-dihydroxy phenyl propionic acid, thereby mitigating macrophage pro-inflammatory activity and increasing the protective effect of nucleotide-biding oligomerization domain-containing protein 2 (NOD2) [62]. This effect is associated with decreased I/R injury, thus improving the early LT outcome. Gut dysbiosis can result in increased bacterial translocation from the gut into the liver allograft, thereby increasing antigen exposure. This can have a dual effect: low-dose antigen exposure in the liver can increase tolerance of the liver towards the allograft, while high-dose antigen exposure can stimulate an enhanced immune response, thereby promoting ACR [63]. 

Patients with gut dysbiosis are more susceptible to negative outcomes, including acute-on-chronic liver failure and advanced cirrhosis in the pre-LT period [64,65]. The impaired microbial functionality in end-stage liver disease is reflected by a lower conversion of bile acids, changes in ammonia metabolism, as well as changes in microbial-mammalian co-metabolites such as trimethylamine-N-oxide (TMAO) [34,66]. Since converted bile acid can suppress pathogens, their reduction becomes worrisome in the pre-LT period [36]. Consequences of exacerbated gut microbial production include hyperammonemia, which can result in the development of hepatic encephalopathy [67]. Studies in animals and cell cultures have shown that TMAO can stimulate the production of various pro-inflammatory molecules, including IL-1β, IL-6, TNF-α, NF-κB, and MMP9, thereby inducing inflammatory responses in the liver and other organs, such as the aortic root [68]. Additionally, TMAO has been shown to link gut microbiota with the development of atherosclerosis and cardiovascular changes [66]. Given the pro-atherogenic profile post-LT, the role of TMAO needs to be further investigated. 

There is evidence especially based on animal trials in rodents, that the immunosuppressive therapy itself induces changes in the gut microbiome [23,69]. Ling et al. report specific alterations in the relative abundance of *Firmicutes* and *Bacteroides* due to tacrolimus treatment in mice [23]. Different effects have been observed for tacrolimus regarding microbial diversity and richness as well the relative abundance of, e.g., *Clostridium*, *Ruminococcaceae*, *Bifidobacterium*, *Bacteroides*, and *Lactobacillus* [69]. Overall, tacrolimus induces a gut dysbiosis comparable to metabolic diseases and alters microbiome-associated metabolic functions such as carbohydrate and lipid metabolism [69]. Mycophenolate Mofetil (MMF), another immunosuppressive drug used in LT, not only causes dysbiosis, which can lead to colitis, but endotoxemia due to an increase in Gram-negative bacteria as well as an impairment of the mucosa barrier and therefore increased gut permeability [69]. Taken together, there are many effects of different used immunosuppressive drugs on the gut microbiome based on animal trials, which may have an impact on LT patients’ outcomes [69]. Only sparse data are available at present, and all these effects are not taken into account in routine clinical practice and treatment of LT patients yet. 

While the current literature on the role of gut microbiota in post-liver transplantation outcomes is limited, studying the role of gut microbiota is crucial. Previous studies have demonstrated a potentially beneficial effect of pre- and probiotics to improve outcomes of post-LT course [70], despite the observed beneficial change in the gut microbiota composition and functionality after successful LT [22]. The effects of necessary immunosuppressive therapy on the gut microbiome and the intestinal barrier have been insufficiently studied to date [69].

## 4. Infections and Colonization, Especially with Multi-Drug Resistant Microbiota

Infections are a major cause of morbidity and even mortality after LT. Despite advances in surgical technique, strategies to prevent infections, and modern immunosuppression regimens, infections still occur in up to 40–50% during short-term and in up to 80% during long-term follow-up after LT [14,15,16].

LT recipients are susceptible to infection due to the technical complexity of the surgical procedure, contamination of the abdominal cavity, and the usually poor medical condition of LT recipients [14,71]. In addition, underlying end-stage liver diseases are related to intestinal and biliary tract dysbiosis; the latter caused by repeated biliary tract interventions [16,72,73]. Recently published data show that end-stage liver disease is associated with immune dysfunction, changes in the local microbial milieu, and an increase in bacterial translocation due to an impaired mucosal barrier [16,72]. Annavajhala et al. demonstrated an inverse correlation of the microbial diversity with MELD- or Child-Pugh score values at the time of transplantation, indicating significant gut dysbiosis at the time of transplantation [16].

The risk for infection is considered to be highest in the first month after transplantation and decreases steadily thereafter [14,71]. Some data report risk factors for post-LT infections, such as patient condition at LT, length of stay in the intensive care unit, prolonged need for catheters, blood transfusions, and duration of surgery [14,15,74,75,76,77]. The most common sites of infection are the abdomen, lungs, bloodstream, and urinary tract [14,15,71,75]. While abdominal infections predominate in the first month in terms of surgical site infections and early bile duct complications, the rate of pulmonary and bloodstream infections increases over time [15,71,75,78].

Most notably, biliary tract infections or cholangitis occur due to leakage and strictures in the bile ducts and are associated with repeated procedures such as Endoscopic Retrograde Cholangiopancreatography (ERCP) [73]. This favors colonization of the bile ducts with various potentially pathogenic germs from the intestine [13,73]. In an analysis by Kabar et al. of bile duct samples from ERCP in LT recipients, 86.6% of samples tested positive for at least potential pathogenic bacteria, and nearly 80% were polymicrobial [73].

Overall, most infections after LT are primarily caused by bacteria, followed by viral and fungal causes [14,15,71,78]. For bacterial infections, in the Gram-negative spectrum, *E. coli*, *Klebsiella* spp., and other *Enterobacteriaceae* are the most important germs, whereas, for Gram-positive germs, *Enterococcus* spp. and *Staphylococcus* spp. are the most common infectious agents [14,15,75,78]. Several studies have reported that especially *Enterococcus* spp. are the predominant pathogens found in LT-related infections [75,76,77]. 

Kim et al. reported 112 episodes of bloodstream infections in 64 LT recipients with *Enterobacteriaceae* spp. (32.5%), *Enterococci* spp. (17.8%), *Staphylococci* spp. (10.3%), and *Acinetobacter baumanii* (10.3%) being the most common germs due to biliary tract and other abdominal or catheter infections [15]. They also note that most germs showed resistance to major antibiotics [15]. Infections caused by multidrug-resistant (MDR) germs are a major problem and are associated with an increased risk for mortality [16,74,79,80]. Several data show that infections with vancomycin-resistant *Enterococcus* spp. (VRE) in particular, are associated with prior antibiotic use, higher rates of biliary complications, more abdominal surgeries, and, most importantly, lower survival [80]. In contrast, colonization with VRE was associated with only a small 1-year mortality risk of about 7% [79]. On the other hand, there is an increasing number of Gram-negative multidrug-resistant germs such as *E. coli*, *Klebsiella* spp., *Citrobacter* spp., and other *Enterobacteriaceae* [74]. Especially in biliary tract and pulmonary infections, these Gram-negative MDR bacteria play an important role by increasing mortality and days in the intensive care unit compared to “normal” non-MDR infections [74].

On the other hand, the risk for infections after LT is determined by the intensity of immunosuppression [31,81]. Especially in the first months after LT, the intensity of immunosuppressive therapy is much more intensive compared to the later immunologically stable course, associated with a correspondingly higher risk of infection [81,82]. Overall, there is a higher susceptibility to bacterial infections due to colonizing potential pathogen germs and opportunistic infections related to immunosuppressive therapy, which can cause serious morbidity [14,81]. Usually, the risk for at least opportunistic infections is addressed by antibiotic prophylaxis in immunosuppressed patients [14,81]. Specifically, for pulses of steroids in acute graft rejection, there seems to be no elevated risk for infections [14].

A recent analysis of the gut microbiome in LT recipients showed that 65% of patients developed colonization with MDR bacteria, e.g., VRE, carbapenem-resistant *Enterobacteriaceae*, or *Enterobacteriaceae* resistant to third generation cephalosporins, within 1 year of LT [16]. Patients colonized with MDR bacteria presented a lower α-diversity throughout the study period, and additional antibiotic exposure significantly decreased gut microbial diversity [16]. However, although most MDR bacterial colonization is known in LT candidates, there are currently no clinical data on the adaption of perioperative antibiotics to the screening results in LT. Adjusting perioperative antibiotic therapy in LT patients for colonization with MDR bacteria could improve perioperative outcomes in terms of severe abdominal, biliary tract, and pulmonary infections.

Given the high rate of infectious complications in LT patients and the associated morbidity and mortality, a better understanding of microbial niches as potential sources of these infections is needed. MDR germs remain a relevant problem, and only homeostasis of the microbial environment and immune function could potentially prevent MDR colonization and associated severe morbidity and mortality.

## 5. Biliary Complications of Liver Transplantation

Biliary complications remain to be the Achilles heel of LT, owing to both surgical and non-surgical factors. Surgical factors include anatomical factors like small bile duct diameter in the graft [83], multiple bile duct orifices [84], intimal damage to the duct, scar formation as a healing process, and compromised blood supply to the bile duct [7,85]. Non-surgical risk factors include increased cold ischemic time [86], arterial hypoperfusion caused by portal hypertension [87], and immunological factors [88]. The overall incidence of biliary complication in LT recipients ranges from 7.4% to 39%; biliary leakage occurs in 5.1–23.4%, and biliary strictures occur in 6.5–21.5% [5,6,7,8]. Biliary complications affect the quality of life in the recipient, leading to significant morbidity and mortality.

### 5.1. Biliary Reconstruction

To minimize the risk of biliary complication in LT, the maintenance of fundamental principles of surgical anastomoses, such as minimal tension, regular intervals between suture bites, sufficient blood supply, and avoidance of injury to bile duct epithelium is of utmost importance [89]. The two most commonly used techniques are the choledocho-choledochostomy or duct-to-duct anastomosis and the choledocho-jejunostomy or Roux-en-Y hepaticojejunostomy. 

Roux-en-Y hepaticojejunostomy has the ability to maintain good blood supply and obtain tension-free anastomosis, which is why it has been promoted in the past [90]. However, duct-to-duct anastomosis is considered more favorable generally because it most closely resembles normal anatomy, thereby eliminating bowel manipulation and preserving the physiological bilio-enteric continuity [91]. Additionally, shorter operation time and easier endoscopic access to the biliary tract in case of biliary complications also make duct-to-duct anastomosis a preferred technique [92]. The rate of biliary complications is either similar between duct-to-duct and Roux-en-Y hepaticojejunostomy or slightly increased for the latter [93]. Colonization of the biliary tract and severe bile leakage and bleeding are more common in Roux-en-Y hepaticojejunostomy [93]. In addition, the infection rate is much higher in Roux-en-Y hepaticojejunostomy (65.9% vs. 22.8%) due to the absence of the sphincter of Oddi, which facilitates ascending bacterial migration and recurrent cholangitis [93,94,95].

### 5.2. Types of Biliary Complications

Biliary leakage and strictures are the most common complications after LT. Leakage mostly occurs from the site of anastomosis and seldom from the T-tube exit side [96]. Anastomotic leaks usually occur within 1 month after LT [97,98]. 

In other gastrointestinal surgeries, the impact of local microbiota causing anastomotic leakage has been unraveled recently [99,100,101]. Despite ischemia and tension at the anastomotic site, bacteria-induced local protease activity can impair anastomotic healing by breaking down newly synthesized collagen [99,101]. Alverdy et al. could prove this impact of certain bacterial strains on enhancing the activity of tissue proteases, e.g., for *enterococci* spp. [100]. In the local microenvironment, commensal bacteria physiologically take part in the modulation of host genes and maintaining the mucosal barrier integrity [99,100]. Surgical trauma, antibiotic use, and states of disease can alter this commensal microbial niche with an increase of at least potential pathogen germs [99,100]. Therefore, the local microbiota might play an important role in the molecular process of anastomotic healing, even in the biliary tract.

Strictures at the site of the biliary anastomosis are relatively frequent and occur in 5% to 10% of patients after LT [102]. The majority of anastomotic strictures happen within the first year after LT [95]. A slight and temporary narrowing at the site of biliary anastomosis after LT is considered normal due to postoperative edema; however, it could further develop into significant anastomotic stricture with relevant cholestasis and cholangitis [103]. The causes of an anastomotic stricture include surgical technique, inadequate mucosa-to-mucosa anastomosis, local ischemia, and fibrotic healing [97]. Generalized ischemia and bile leakage also increase the risk of anastomotic structuring. Most anastomotic strictures are treatable with endoscopic procedures with high success rates. 

Non-anastomotic strictures (NAS) occurring after hepatic artery thrombosis has been well-known since the beginning of liver transplantation [104]. NAS occurring with a preserved arterial blood supply has been described as ischemic-type biliary lesions (ITBL) in the 1990s and represents a major therapeutic problem [105]. ITBL is associated with the destruction of the non-anastomotic parts of the biliary tract, including segmental stenosis and expansion, resulting in biliary sludge, biliary casts, and filling defects [9]. The development of ITBL is associated with significant morbidity due to the need for multiple biliary interventions, and approximately 65% of patients with ITBL require re-LT [18]. In general, rates of NAS are up to 19%, and rates of ITBL range from 3% to 16% following LT [17]. An ischemic/immune-mediated injury is the most straightforward pathogenesis of NAS; in addition, surgical factors and cytotoxicity of bile salts may also play a role in the development of NAS [88,106]. Damage of the bile duct epithelium or injury to the microvasculature of the bile duct arteriolar plexus due to fibrotic healing could lead to these strictures [107]. Identified risk factors for the development of NAS include macro-angiopathic (hepatic artery thrombosis or stenosis), microangiopathic (prolonged ischemia times, preservation solution, cardiac death donor, donor dopamine use), and immunogenetic (ABO incompatibility, rejection, auto-immune disease, CMV infection, chemokine polymorphisms) injury [108]. Another potential factor in the pathogenesis of bile duct injury after LT is the local microbiota, as a biliary infection is a frequent cause of biliary complications in patients after LT [30,31]. Recently, a study suggested enteric bacteria to be significantly associated with the clinical signs of cholangitis after LT, a condition that lowers survival rates in patients with biliary tract injury [32].

Microbiome research of the gut, but especially the biliary microbiome, might contribute to a deeper understanding of the pathomechanisms of biliary complications, especially biliary leak and anastomotic and non-anastomotic strictures. By influencing microbial niches, new therapeutic and even prophylactic options may become available in the future.

## 6. Microbiota in Liver Transplantation and Associated Biliary Complications

The physiologic colonization of the gut and the biliary tract changes with the occurrence of liver diseases. In these instances, pathogenic potentially harmful bacteria can be found, which are then summarized under the term “pathobiome” [26], whereas some microbes might exhibit beneficial effects against the development of liver diseases [22]. Table 1 provides an overview of these relevant microbiota. A growing number of studies have begun to elucidate the role of the microbiota, its metabolites, and its influence on host immune responses after LT in general and specifically in the development of biliary complications. Various factors from microbiota potentially contributing to the development of biliary complications are summarized in Figure 2.

### 6.1. Gut Microbiota

Gut bacteria have a significant impact on human metabolic activity, barrier function, and immunity development. Dysbiosis of gut bacteria is associated with various conditions, including obesity, diabetes, nonalcoholic fatty liver diseases, and autoimmune disorders [121,122], and even plays a significant role in I/R injury [123]. For LT patients, portal vein blocking, I/R injury, antibiotics, or immunosuppression use can seriously impair the recipient’s immune function, and destroy the intestinal barrier, thereby increasing the risk of dysbiosis of gut microbiota. These changes in gut bacteria may lead to direct injury to the host liver through the “gut-liver axis” [124]. The relationship between gut microbiota dysbiosis and postoperative complications, including acute rejection, early-stage infection, and graft loss due to biliary complications, has been discussed in the previous sections. Based on the sparse available data, one can hypothesize that alterations of gut microbiota may be responsible for the graft’s I/R injury to some extent.

Compared to patients without complications after LT, patients diagnosed with NAS showed a decreased abundance of *Bacteroidetes* and an increase of *Proteobacteria,* which usually amounts to a very small part of human gut microbiota [113,125]. *Bacteroides*, together with *Firmicutes*, are predominant bacteria within the human intestine; a decrease of either always indicates an impairment of intestinal barrier function and an increased risk of bacterial translocation [126]. Similar changes have also been reported in cirrhosis patients [47]. At the family level, higher proportions of *Enterococcaceae*, *Streptococcaceae*, *Enterobacteriaceae*, and *Pseudomonadaceae* were observed among patients with NAS in comparison with patients without biliary complications of LT [113]. These families of bacteria are commonly regarded as pathogenic bacteria, and their overgrowth will lead to a release of LPS and peptidoglycan. When recognized by the human immune system via Toll-like receptors or nucleotide-binding oligomerization domain-like receptors, LPS, and peptidoglycan would trigger the pro-inflammatory NF-κB cascade and directly stimulate hepatic stellate cells, which finally lead to liver damage and liver disease progression [124]. As bile ducts are susceptible to inflammatory damage, serious gut microbiota dysbiosis may exacerbate cholangiocyte apoptosis and eventually lead to bile duct strictures or ITBL [108]. In addition, cholangiocytes are also susceptible to ischemic injury and oxygen-free radicals, and microcirculatory disturbances can lead to insufficient biliary tract preservation as it is caused by I/R injury, which involves inflammation, oxidative stress, apoptosis, and necrosis [127]. The production of reactive oxygen species influenced by microbiota could further induce cholangiocyte-related damage [128,129].

### 6.2. Biliary Microbiota

The biliary tract is not sterile despite the anti-microbial activity of bile acids, even in healthy individuals [26,27]. In certain diseases or infectious conditions, reduced bile acid secretion can increase bacterial biliary colonization [130]. Especially in diseases of the biliary tract, e.g., PSC, studies revealed an altered microbial niche [11]. This dysbiosis is associated with an increased pro-inflammatory and even carcinogenic milieu [11,12,131]. Especially, *Proteobacteria* and *Enterococci* spp. are enriched in PSC and other end-stage liver diseases [11,131]. Furthermore, endoscopic interventions, biliary stents, and recurrent antibiotic therapy could alter the microbiota within the biliary tract, including ascending bacteria from the upper intestines [132].

Only sparse data are available on the biliary microbiome in different diseases and especially in LT patients [10,11,12,13]. Despite the study of D’Amico et al., who could not detect any biliary microbiome in a small series of six patients [133], data are mostly available for bile samples of patients suffering from biliary complications. Liu et al. analyzed bile samples from liver transplant recipients with routine use of biliary drains [10]. The predominant bacterial phyla were *Firmicutes*, *Proteobacteria,* and *Actinobacteria*, with differences in relative abundance between patients with and without biliary complications such as cholangitis or stenosis [10]. Recently, Klein et al. reported on a large series of patients using 16S rRNA-based microbiome analysis on bile samples [13]. They compared the biliary microbiome of patients with non-anastomotic vs. anastomotic strictures as controls. They detected a diverse biliary microbiome consisting mainly of the phyla *Firmicutes*, *Proteobacteria*, *Fusobacteria*, *Bacteroidetes,* and *Actinobacteria,* with differences in relative abundance between the groups [13]. On the Genus level, especially *Enterococci* spp. and *Streptococci* spp. have been found in all samples [13]. The microbial community structures were different between groups with biliary stents and recent antibiotic therapies [13]. Whereas biliary stenting did not result in different abundance in the anastomotic stricture group, the biliary microbiome differed in the non-anastomotic stricture group: they detected differences in the relative abundance of 27 genera in the microbial community of samples with and without biliary stents [13]. Despite an increase of biofilm-forming bacteria over time with the use of biliary stents, e.g., *Streptococci* spp. and *Fusobacteria* spp., the analysis revealed no relevant difference in diversity and similarity in the non-anastomotic stricture group due to antibiotic therapy [13].

Overall, the sparse available data demonstrates an increase of *Proteobacteria* in bile samples of patients with biliary complications, indicating an increase in potentially pathogen germs like *E. coli*, *Klebsiella,* and other Gram-negative germs. This was consistent with the gut microbiome in NAS discussed before, suggesting that *Proteobacteria* may play a significant role in the occurrence and development of biliary tract injury after LT. If the increase of *Proteobacteria* and these differences in relative abundance are causes or consequences of the biliary complications still remains unclear.

However, unlike the gut microbiome, in patients with NAS, the proportion of *Enterococcus* spp. in bile was lower than in patients without complications [10]. This bacterium can regulate the balance of intestinal flora and process certain immune regulatory and anti-allergic effects [134]. Reduction in the biliary abundance of *Enterococcus* spp. in NAS patients suggests that *Enterococcus* spp. may play an important role in maintaining the stability and balance of bile microecology.

The impact of biliary microbiota on biliary complications like anastomotic strictures or ITBL cannot be proven in prospective sampling studies, but at least due to cholestasis-related cholangitis and especially repeated endoscopic interventions like ERC and application of biliary drainages, there is an increase in potentially pathogen microbiota like *Proteobacteria* and *Enterococci* spp. [10,13,32]. Due to repeated interventions in patients suffering from biliary complications and the need for antibiotic therapy even before LT, the colonization and associated complications due to multi-drug resistant microbiota are increased [13,16,32]. In the available data, there were some significant differences in the metabolic pathways of bile samples between patients with and without biliary complications [10,13]; however, there are limited data on metabolic pathways in bile samples from patients after LT, and therefore further data are required.

There are complex interactions between the gut microbiome and liver function as well as immune regulation with a significant impact on outcome in liver transplantation. Notably, there are only sparse data on the microbial niche of the biliary tract in LT. A more detailed description of the biliary microbiome is required both in the physiological state and in various biliary and immunological complications of liver transplantation. Table 1 provides an overview of the current knowledge of various microbiota detected in gut and biliary samples from patients with (end-stage) liver diseases or after LT.

## 7. Microbiota as a Predictive Tool and Therapeutic Target

Biliary complications, as well as graft function in LT, are influenced by many different factors. The microbiome within the human gut and biliary tract plays an important role in the pathogenesis and development of complications after LT. Hence, microbiota presents itself as a very useful predictive tool for post-LT outcomes in data from animal experiments. For example, a murine model could differentiate the cause of liver dysfunction by using gut microbial profiling. As such, fecal microbiota sampling can serve as a potentially better biomarker for early detection of the various post-LT complications due to its non-invasive nature [135]. Data regarding the role of the microbiome in hepatobiliary disease and LT mostly aim at the human gut and fecal microbiome profiling, showing specific pathogenic alterations in the feces [26,113,136]. In terms of biliary microbiota, the data is scarce, and the only study related to biliary complications in LT patients showed there were significant differences in species composition within the bile samples between patients with and without biliary complications after LT [10].

It is highly advantageous if the rise of certain species or pathways within biliary and fecal samples could be associated with certain complications. This prompts the consideration of therapeutic alteration of microbiota composition. Since the liver is an immunotolerant organ, the discontinuation of the immunosuppression may be capable [137]. Therapeutic alteration of the microbiome could be considered, thereby preventing or improving post-LT complications. Such therapeutic alteration can be achieved through the administration of probiotics or fecal microbiota transplant. Existing literature on probiotic usage in patients after LT suggests efficacy in reducing post-LT infection [138], and fecal microbiota transplants have been demonstrated to be beneficial toward alcohol-related liver disease and hepatic encephalopathy [139,140]. However, there is no data available at present regarding biliary complications after LT.

## 8. Conclusions

Microbiome research can be a useful aid in LT patient care; however, the current understanding of the roles of microbiota in biliary complications and graft functions in LT is inadequate for clinical application due to small sample sizes and the limited data from human studies. Future studies should not only focus on the composition and diversity of microbiota. Specific microbial characteristics such as metabolomics and transcriptomics should also be taken into consideration. The same microbiota may exhibit different behaviors in different contexts. Especially data on the biliary microbiome is still sparse, and their impact on infectious and especially biliary complications is mainly speculative. The effects of immunosuppressive therapy following LT on the gut and even biliary microbiome have not been sufficiently analyzed yet; in particular, this aspect does not currently play a role in the daily clinical care of our LT patients. Further data from prospective clinical trials are necessary for a better understanding of these complex interrelationships. Since most LT complications could be acute or chronic, an adequate follow-up period with a collection of bio-samples is also required.

## Figures and Tables

**Figure 1 ijms-24-04841-f001:**
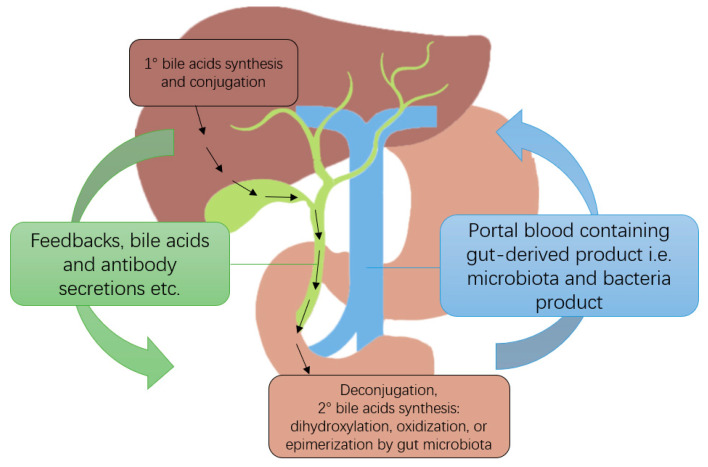
Illustration of the gut-liver axis and bile acid metabolism.

**Figure 2 ijms-24-04841-f002:**
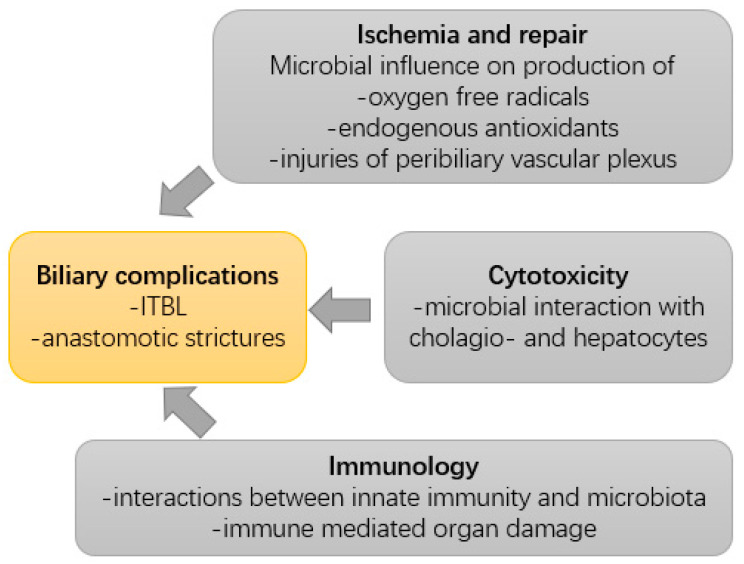
Multifactorial pathogenesis of biliary complications with a focus on potential microbial causes.

**Table 1 ijms-24-04841-t001:** Overview of the microbiota associated with (end-stage) liver diseases as well as biliary complications in liver transplantation patients (PSC: primary sclerosing cholangitis; LT: liver transplantation; NAS: non-anastomotic strictures).

Phylum	Family	Genus	Characteristics
Actinobacteria	*Micrococcaceae*	*Rothia*	Part of the oral microbiota. Increased abundance within the gut during PSC [109]. Contamination from endoscopic procedures is suspected [109].
	*Propionibacteriaceae*	*Cutibacterium*	Prevalent on human skin and frequently associated with skin conditions. Increased abundance in the biliary tract in patients with PSC suspected to be contamination from endoscopic procedures [110].
Bacteroidetes	*Bacteroidaceae*	*Bacteroides*	Frequent colonization of the human gut. Increased amount in the gut is found among PSC patients [111], whereas its proportion decreases in advanced cirrhosis patients [34]).
	*Prevotellaceae*	*Prevotella*	Produce butyrate, which promotes intestinal barrier function. In PSC patients, abundance is decreased in the gut [109,111] but increased in the biliary tract [110].
	*Tannerellaceae*	*Parabacteroides*	Frequent colonization of the human gut. Increased amount in the gut is found among patients with PSC [112]
	*Staphylococcaceae*	*Staphylococcus*	Frequent colonization of human skin, nasopharynx, and gut. Therefore, its increased abundance in the biliary tract for patients with cholangitis and other liver diseases is mostly due to contamination from endoscopic procedures [110].
Firmicutes	*Enterococcaceae*	*Enterococcus*	Frequent colonization of the gut and biliary tract. Its abundance is associated with an increase of taurolithocholic acid, a proinflammatory and cancerogenic type of bile acid, therefore frequently associated with cholangitis [110]. Increased abundance in patients with PSC both in the biliary tract and the gut [109,110,112]. In patients with biliary complications, its abundance is increased in the gut [113] but higher in patients with NAS than in patients with anastomotic stricture [13].
	*Lachnospiraceae*	*Roseburia*	Frequent colonization of the human gut. Can perform 7α-dehydroxylation. Produces primary amines that act as vascular adhesion protein-1, which is critical for effector cell recruitment to the liver [109]. Produces butyrate, which promotes intestinal barrier function. Decreased amount in the gut in PSC patients [109,111]
	*Lactobacillaceae*	*Lactobacillus*	Frequent colonization of the human digestive system and female genital system. Increased colonic colonization is found in patients with PSC and other live diseases [114].
	*Veillonellaceae*	*Veillonella*	Commensal bacteria of human intestines and oral cavity. 70% of the strains are resistant to penicillin [115]. Produces primary amines that act as vascular adhesion protein 1, which is critical for effector cell recruitment to the liver [109]. In patients with PSC, there is increased colonization in the gut and the biliary tract [109,110].
		*Megasphaera*	Can perform 7α-dehydroxylation. Produce primary amines that act as vascular adhesion protein-1, critical for effector cell recruitment to the liver [109]. Increased amount in the gut in PSC [116].
	*Streptococcaceae*	*Streptococcus*	Part of the oral microbiome. Increased amount in the gastrointestinal and biliary tract in PSC patients [109,110,112,116]. In liver LT patients, increased abundance is associated with biliary stent use due to its biofilm-forming nature [13]. It is increased in both the gastrointestinal and biliary tract for patients with NAS [13,113]. Contamination from endoscopic procedures is suspected [26].
	*Clostridiaceae*	*Clostridium*	Play an important role in colonic homeostasis by influencing the Treg and the production of proinflammatory cytokines [11]. Increased amount in the gut is found among PSC patients [111,116], whereas its proportion decreases in advanced cirrhosis patients [34].
Fusobacteriota	*Fusobacteriaceae*	*Fusobacterium*	Increased amount in the gut in PSC patients and associated with CRC cancerogenesis [117]. Increased abundance is associated with biliary stent use due to its biofilm-forming nature [13].
Proteobacteria	*Enterobacteriaceae*	*Escherichia*	Commensal bacteria of the gut microbiota. It can perform 7α-dehydroxylation, which converts primary bile acids to secondary bile acids [11]. It also produces primary amines that act as vascular adhesion protein 1, which is critical for effector cell recruitment to the liver [109]. It is associated with increased colonic epithelial oxygen availability, inflammation, epithelial dysfunction, and disease [118]. It is the predominant pathogen of spontaneous bacterial peritonitis in liver cirrhosis [119]. Increased abundance is found in the gut among patients with NAS [113] and end-stage liver disease [34,110,116].
		*Klebsiella*	Increased in patients with NAS [113]. In addition, it frequently causes spontaneous bacterial peritonitis [119].
	*Pseudomonadaceae*	*Pseudomonas*	A frequent cause of infections in patients with end-stage liver disease and cholangitis [120]. Increased in the gut among patients with NAS [113]
	*Neisseriaceae*	*Neisseria*	Linked to H_2_S production that can damage deoxyribonucleic acid [119]. Increased colonization in the biliary traction during PSC [110,119].
	*Sphingomonadaceae*	*Sphingomonas*	Express amine oxidases and associated with aberrant homing of gut lymphocytes to the liver [111]. Increased amount in the gut in PSC patients [111].

## Data Availability

Not applicable.

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
