# Peer review of "The Role of Microbiota in Liver Transplantation and Liver Transplantation-Related Biliary Complications"

_ijms, 2023, doi:10.3390/ijms24054841_

Round 1

Reviewer 1 Report

The authors sum up the current knowledge of biliary microbiome composition and influence on LT outcome in a very comprehensive and clear way. The Review is easy to follow and provides the reader with a broad range of Information in the field as starting point for in-depth literature research.

Author Response

We thank this reviewer for his comment on our manuscript and revised the manuscript according to the comments and suggestions of reviewer 2 and 3.

Reviewer 2 Report

In this manuscript, the authors well summarized pathological significance of gut and biliary microbiota in liver transplant recipients and complications associated with liver transplantation. They provide useful and interesting information, which are nicely organized throughout the text. Nonetheless, I would like the authors to address my following comments:

Abstract and Introduction: Background information and clinical problems in terms of multi-drug resistant microbiota in the biliary complications of liver transplant patients should be provided.

lines 38–41: Very similar sentences are appeared in lines 64–66. Please revise either one.

lines 106–115: If it has been proved that altered microbiome predicts the prognosis of recipients after liver transplantation, please explain it by citing the paper.

lines 150–158: TMAO enhances anti-cancer immunity by activating effector T cells (Sci Immunol 2022;7:abn0704). Actually, several papers suggesting that TMAO causes inflammatory response in the liver have been reported. This point should be mentioned.

Section 3: The title should be changed from "Microbiota···" to "Gut microbiota···" if there is no report regarding biliary microbiota. to be introduced in this section.

Section 4: The effects of immunosuppressive agents on infectious complications should also be discussed.

Section 6.2: Are there any relationship between biliary microbiota, complications, and infection with drug-resistant bacteria?

Overall, this article is well written and worth to be published. However, I would like to recommend for the authors to consider other more appropriate, clinical journals because the contents are not related to molecular mechanisms and cancers.

Author Response

Abstract and Introduction: Background information and clinical problems in terms of multi-drug resistant microbiota in the biliary complications of liver transplant patients should be provided.

Answer: we added the additional aspects on background information to the abstract and introduction section.

lines 38–41: Very similar sentences are appeared in lines 64–66. Please revise either one.

Answer: we revised the manuscript and changed the latter sentence in (former) lines 64-66.

lines 106–115: If it has been proved that altered microbiome predicts the prognosis of recipients after liver transplantation, please explain it by citing the paper.

Answer: we added some paragraph about impact of microbiome / microbial changed e.g. due to pre-/probiotic therapy on patients’ outcome in liver transplantation. A real prognosis based on microbial signatures is not possible yet due to sparse available data as described in the manuscript, but some data from animal experiments seem to be promising.

lines 150–158: TMAO enhances anti-cancer immunity by activating effector T cells (Sci Immunol 2022;7:abn0704). Actually, several papers suggesting that TMAO causes inflammatory response in the liver have been reported. This point should be mentioned.

Answer: we added the missing information about inflammatory response due to TMAO.

Section 3: The title should be changed from "Microbiota···" to "Gut microbiota···" if there is no report regarding biliary microbiota. to be introduced in this section.

Answer: we changed the section title as recommended.

Section 4: The effects of immunosuppressive agents on infectious complications should also be discussed.

Answer: we added a paragraph about effect of immunosuppressive therapy on infectious complications as well as another paragraph about effects of immunosuppressive therapy on microbial changes in section 3.

Section 6.2: Are there any relationship between biliary microbiota, complications, and infection with drug-resistant bacteria?

Answer: We revised this section and added some specific information about the relationship between biliary microbiota, complications and infections with MDR germs.

Overall, this article is well written and worth to be published. However, I would like to recommend for the authors to consider other more appropriate, clinical journals because the contents are not related to molecular mechanisms and cancers.

Answer: we thank you for your comment, but in our opinion microbiome research is on a molecular basis and the International Journal of Molecular Sciences is not exclusively dealing with cancer. Therefore, we revised our manuscript according to all the reviewers’ comments for possible publication in International Journal of Molecular Sciences.

Reviewer 3 Report

In this article, the authors review the role of microbiota in liver transplantation and liver transplantation-related biliary complications.

My comments are as follows:

The references used are on the whole not very up to date

 "Recent studies 96 have demonstrated the essential role of innate immune responses" This sentence is highlighted.

Line 138, ischemia-reperfusion damage is mentioned for the first time when it should be dealt with in great depth in this article because it is one of the major problems in liver surgery.

Please specify whether the studies named throughout the text are done in animals or in humans

I believe that the use of antibiotics should be included in a separate section.

The conclusion/discussion of this paper seems poor to me.

When talking about liver transplantation, what is the cause? It is not the same for a patient with cancer as for a cirrhotic patient.

Readers should be informed more about the evolution of the liver disease.

Author Response

The references used are on the whole not very up to date

Answer: we checked the references and added some up to date sources, but overall the data on microbiome in LT, infectious complications are sparse and therefore, we need to refer to some older references as well.

 "Recent studies 96 have demonstrated the essential role of innate immune responses" This sentence is highlighted.

 Answer: we changed the formatting of the above mentioned sentence.

Line 138, ischemia-reperfusion damage is mentioned for the first time when it should be dealt with in great depth in this article because it is one of the major problems in liver surgery.

Answer: We agree that ischemia reperfusion damage is a major problem in liver transplantation, but only sparse data regarding the relevance of the microbiome are available at present and we presented them in the above mentioned paragraph as e.g. there are interactions with the innate immune system and effects and microbial SCFA metabolism. Therefore, we are not able to much extend this paragraph / section.

Please specify whether the studies named throughout the text are done in animals or in humans

Answer: we specified the relevant information, if the mentioned studies were performed in humans or animals.

I believe that the use of antibiotics should be included in a separate section.

Answer: We thank the reviewer for this suggestion. There are a lot of data dealing with antibiotic effects on the gut microbiome, but such a discourse would go beyond the scope of this manuscript. We added some information about perioperative effects of applicated broad spectrum antibiotics in liver transplantation and already included some aspects of repeated use of antibiotics in pre-/post liver transplantation in infections with risk for selection of multi-drug resistant germs.

The conclusion/discussion of this paper seems poor to me.

Answer: as all important aspects of gut / biliary microbiome in liver transplantation and biliary complications have been comprehensively reviewed, we only added a short conclusion. We now revised the conclusion and hope to have upgraded it.

When talking about liver transplantation, what is the cause? It is not the same for a patient with cancer as for a cirrhotic patient.

Answer: We thank the reviewer for this comment. The focus of our work is the role of microbiota in liver transplantation. There are different works on different diseases leading to end-stage liver disease, but a comprehensive review of all important aspects for all different underlaying diagnosis was not intended and seems to exceed the goal of this work. We already included some known aspects of different end-stage liver diseases, especially PSC as there are a lot of microbiome data available. We added some information based on the work of Annavajhala et al. (Nat Com doi:10.1038/s41467-019-12633-4), demonstrating that the state of the liver disease and different underlaying diagnosis can be distinguished by gut microbiome analysis. Furthermore, the relevant microbiota in different liver diseases and liver transplantation are summarized in Table 1.

Readers should be informed more about the evolution of the liver disease.

Answer: Our review is focusing on microbiome in liver transplantation with focus on perioperative infectious complications and short as well as long term biliary complications. To discuss the evolution of different liver diseases will be the aim of another review, but this recommendation is beyond the scope of this manuscript.

Reviewer 4 Report

This review has very sincere conclusions on the role of the microbiota in liver transplantation, biliary complications, and its scope in clinical practice.

Some minor aspects could substantially improve this review:

1. The abstract is not very informative of what was developed by the authors in the review.

2. line 15. The authors should not rule out the higher mortality risks after liver transplantation in chronic graft rejection.

3. Some types are observed throughout the document, in the manuscript, figure, and tables: line 31, 285, fig2…” strictures”.

4. Line 83. The complete genus of the microorganism must be indicated.

5. line 109. References to the said statement are missing.

6. line 149. Explain this statement better.

7. The entire document should be grammatically checked.

8. Some references could be incorporated into this review.

https://doi.org/10.1111/acer.13013

https://doi.org/10.3390/diagnostics11060968

https://doi.org/10.3390/ijms232012155

Author Response

  1. The abstract is not very informative of what was developed by the authors in the review.

Answer: we revised the abstract and extended it especially with more background information.

  1. line 15. The authors should not rule out the higher mortality risks after liver transplantation in chronic graft rejection.

Answer: we take this aspect into account in the abstract as well as the review itself.

  1. Some types are observed throughout the document, in the manuscript, figure, and tables: line 31, 285, fig2…” strictures”.

Answer: we revised the language and grammar of the review, but especially the word “stricture / strictures” seemed to be correct in our opinion.

  1. Line 83. The complete genus of the microorganism must be indicated.

Answer: we name the complete genus of the microorganism now.

  1. line 109. References to the said statement are missing.

Answer: We added the relevant references.

  1. line 149. Explain this statement better.

Answer: we extended this statement with an informative explanation.

  1. The entire document should be grammatically checked.

Answer: we checked the grammar and language of the whole manuscript.

  1. Some references could be incorporated into this review.

https://doi.org/10.1111/acer.13013

https://doi.org/10.3390/diagnostics11060968

https://doi.org/10.3390/ijms232012155

Answer: we screened the suggested references and included two of them into the review, as they harbor interesting details, which we now report of. We thank the reviewer for this recommendation.

Round 2

Reviewer 2 Report

I appreciate the authors for sincerely addressing my comments. The manuscript has been substantially improved.

Reviewer 3 Report

The authors have not answered the vast majority of my questions and comments which I consider important to put the work in context.

However, in view of the reports of the other reviewers, I will not reject it and leave it to the editor to decide whether the authors should include them or not.